# Green synthesis and characterization of silver nanoparticles and its efficacy against *Rhizoctonia solani*, a fungus causing sheath blight disease in rice

A. K. M. Sahfiqul Islam[1], Rejwan Bhuiyan[1], Sheikh Arafat Islam Nihad[1], Rumana Akter[1], Mohammad Ashik Iqbal Khan[1], Shamima Akter[1], Md. Rashidul Islam[2], Md. Atiqur Rahman Khokon[1], Mohammad Abdul Latif[1] *

1 Plant Pathology Division, Bangladesh Rice Research Institute (BRRI), Gazipur, Bangladesh, 2 Department of Plant Pathology Division, Bangladesh Agricultural University, Mymensingh, Bangladesh

* alatif1965@yahoo.com, latifbrri@gmail.com

**Data Availability Statement:** All relevant data are within the paper and its Supporting Information files.

## Abstract

Rice (*Oryza sativa*) stands as a crucial staple food worldwide, especially in Bangladesh, where it ranks as the third-largest producer. However, intensified cultivation has made high-yielding rice varieties susceptible to various biotic stresses, notably sheath blight caused by *Rhizoctonia solani*, which inflicts significant yield losses annually. Traditional fungicides, though effective, pose environmental and health risks. To address this, nanotechnology emerges as a promising avenue, leveraging the antimicrobial properties of nanoparticles like silver nanoparticles (AgNPs). This study explored the green synthesis of AgNPs using *Ipomoea carnea* leaf extract and silver nitrate ($AgNO_3$), and also examined their efficacy against sheath blight disease in rice. The biosynthesized AgNPs were characterized through various analytical techniques such as UV-vis spectrophotometer, X-ray Diffraction (XRD), Fourier Transform Infrared Spectroscopy (FTIR), Particle size analyzer, Zeta potential, Scanning Electron Microscope (SEM), Field Emission Scanning Electron Microscope (FESEM), Transmission Electron Microscope (TEM) for confirming their successful production and crystalline nature of nanoparticles. The results of UV-visible spectrophotometers revealed an absorption peak ranging from 421 to 434 nm, validated the synthesis of AgNPs in the solution. XRD, DLS, and TEM estimated AgNPs sizes were ~45 nm, 66.2nm, and 46.38 to 73.81 nm, respectively. SEM and FESEM demonstrated that the synthesized AgNPs were spherical in shape. *In vitro* assays demonstrated the significant inhibitory effects of AgNPs on mycelial growth of *Rhizoctonia solani*, particularly at higher concentrations and pH levels. Further greenhouse and field experiments validated the antifungal efficacy of AgNPs against sheath blight disease in rice, exhibiting comparable effectiveness to commercial fungicides. The findings highlight the potential of AgNPs as a sustainable and effective alternative for managing rice sheath blight disease, offering a safer solution amidst environmental concerns associated with conventional fungicides.

**Funding:** This experiment was conducted under the project titled "Sustainable management of blast, sheath blight and bacterial blight diseases of rice through Nanoparticles (Project ID:TF 71-C/20) funded by Krishi Gobeshona Foundation (KGF), Ministry of Agriculture, Bangladesh. The funders had no role in study design, data collection and analysis, decision to publish, or preparation of the manuscript.

**Competing interests:** The authors have declared that no competing interests exist.

## 1. Introduction

Rice (*Oryza sativa*) is an ancient and one of the most ideal staple foods for human beings. The wide range of adaptability to different ecosystems from hilly areas to submergence makes it more acceptable over other crops. Even if diets are becoming more diverse these days, rice still accounts for over half of the world's daily caloric intake and is therefore the most important staple meal for over half of the human population [1, 2]. Research indicates that the worldwide population is expected to rise by 7 to 10 billion people by 2050, leading to a significant rise in the demand for rice, even with the assurance that rice yields will increase by 25% by 2030 [2]. Bangladesh is a large contributor to global rice production and it ranked as the third rice-producing country in the world. Over the last 30 years, rice production in Bangladesh has significantly increased across the country [3]. Intensive rice cultivation and high input requirements make the high-yielding varieties vulnerable to different biotic stresses [4]. So far 32 rice diseases have been identified in Bangladesh, and among them blast, sheath blight, bacterial blight, false smut and tungro are the major diseases of rice in the country [5–9].

Sheath blight caused by *Rhizoctonia solani* signifies its importance due to its severity and lack of resistant source in the country as well as in the world [9]. Sheath blight of rice is a very destructive disease that results in quality degradation and up to a 50% yield loss each year throughout the world [10–12]. Only a small number of fungicides are available for the management of rice sheath blight, and chemical control is the primary method of disease management [13]. Numerous fungicides, including benomyl, captafol, carbendazim, carboxy, chloroneb, edifenphos, mancozeb, thiophanate, and zineb, were successful in controlling the sheath blight disease in the field [14, 15]. But the excessive and mischievous use of these fungicides could result in developing resistance to the pathogen [16] and cause long-term harm to the environment and human health [17]. However, to ensure environmental safety there is a need for new fungicides with less toxicity, high selectivity, and high activity against fungal strains that are resistant to other fungicides. The approach to the discovery of new fungicides and new formulations against rice sheath blight is of vital importance.

Nanotechnology represents a possible new management tool since nanoparticles have been reported to have microbial inhibitory effects [18]. Nanoparticles are characterized by their particle size, which generally ranges between 1 to 100 nm [19, 20]. In recent years, the use of nanomaterials has had a significant impact on several high-tech industries, including medication delivery, cancer therapy [21], energy and biomedicine [22] agriculture [23], and many other sectors [24]. Using atomic-scale materials, new methods of disease control have been made possible by nanotechnology [25]. Enormous surface-to-volume ratio offers a large contact surface with pathogen sources, very small-scale particles have emerged as contemporary agents [26]. Small-scale tools [27], plant protection products [28], fertilizers [29], water purification and pollutant remediation [30], nano sensors, diagnostic tools [31], and plant genetic modification [32] are just a few examples of how nanotechnology can have a significant impact on natural processes and agriculture.

It is well known that silver nanoparticles (AgNPs) are widely used in fields such as photonics, catalysis, bio-nanomaterials, and medicine due to their inherent morphology, composition, and crystal structure [33–35]. AgNPs are highly reactive and have potential inhibitory effects against bacteria and fungi [36, 37]. They can absorb quickly into cells, even at low concentrations resulting death of that microorganism [37]. In large-scale biological processes, AgNPs can damage microorganisms, including the cell membrane structure [38, 39]. The basis for the antibacterial effect of silver ions is to pass through the bacterial cell wall and to alter cellular signaling [40]. AgNPs with fungistatic, bacteriostatic, and plasmonic characteristics are among the more environmentally friendly plant pathogen inhibitors when compared to

synthetic fungicides [41]. AgNPs inhibit the mycelium growth of phytopathogenic fungi including *Alternaria alternata*, *Botrytis cinerea*, and *Colletotrichum gloeosporioides* [42]. Therefore, the antifungal activity of AgNPs has drawn less attention than it has in the medical and pharmaceutical sciences.

At present, the preparation methods used for AgNPs are mainly physical, chemical, and/or biological methods [43]. The wide dispersion, high purity, and strong activity of the nano-silver particles produced by physical methods come at the cost of using high pressure and temperature conditions during the procedure. Because of the high energy consumption and the tendency of the produced particles to aggregate, AgNPs operate less effectively [44]. Although the chemical method can yield AgNPs with a narrow particle size distribution range and difficult to agglomerate, the reaction process employs strong reducing agents such as hydrazine, dimethylformamide, sodium borohydride, and other toxic organic reagents, which pose a serious threat to the environment [45, 46].

In recent years, due to the increasing concern towards the subject of green chemistry, by virtue of it being simple, efficient, and pollution-free, the biological method with microorganisms and plants has attracted the interest of researchers [47]. Due to their ecologically benign character, plant systems have been utilized as a dependable, green technique for the production of nanoparticles [43]. This method is easier to use, employs mild conditions, takes less time to complete, and has easy access to ingredients, making it more ideal for the production of AgNPs [45]. It has been reported that the plant materials can be used to synthesize AgNPs, such as *Ficus carica*, *Coffea arabica*, *Rosmarinus officinlis*, *Lawsonia inermis*, *Aloe vera*, *Eclipta alba*, *Momordica charantia*, *Leptadenia reticulataand* and so on [48, 49].

Considering the advantage of green nanoparticles, this study was undertaken to synthesize silver nanoparticles using $AgNO_3$ and *I. carnea* (Dholkolmi) leaf extract. To confirm and characterize the AgNPs, different methods were applied in this experiment. Finally, we conducted *in vitro* and *in vivo* experiments to test the efficacy of AgNPs against *R. solani* causing sheath blight disease in rice.

## 2. Materials and methods

### 2.1. Preparation of leaf extracts

Dholkolmi (*Ipomoea carnea*) leaf was used as a reducing agent to biosynthesize silver nanoparticles (AgNPs). This plant was selected due to its low cost, convenience of availability, and therapeutic qualities. From the territory of the Bangladesh Rice Research Institute (BRRI) in Gazipur-1701, fresh Dholkolmi leaves were collected. These were properly cleaned under running water to get rid of dirt and other organic impurities, and then washed a couple of times with distilled water, and let air dry at room temperature. A beaker containing 20 g of finely chopped leaves and 300 mL of distilled water was heated to 80˚C and then boiled for 30 minutes. The extract was cooled down and twice filtered using Whatman No. 1 filter paper to get rid of contaminants and to get clean solutions.

### 2.2. Biosynthesis of AgNPs

Silver nitrate ($AgNO_3$) solution with 2 mM concentration was prepared for the manufacture of AgNPs. For the green synthesis of AgNPs, three distinct leaf extract concentrations (1 mL, 2 mL, and 3 mL) were used in these studies. Leaf extract solutions were mixed with 10 mL solution of 2 mM $AgNO_3$ at three different ratios viz. 1:10, 2:10, and 3:10 (Dholkolmi leaves extract (D):$AgNO_3$). After mixing, a distinct color transformation (from light orange to dark brown) was seen, indicating the biosynthesis of AgNPs. Effect of pH was studied by changing the pH

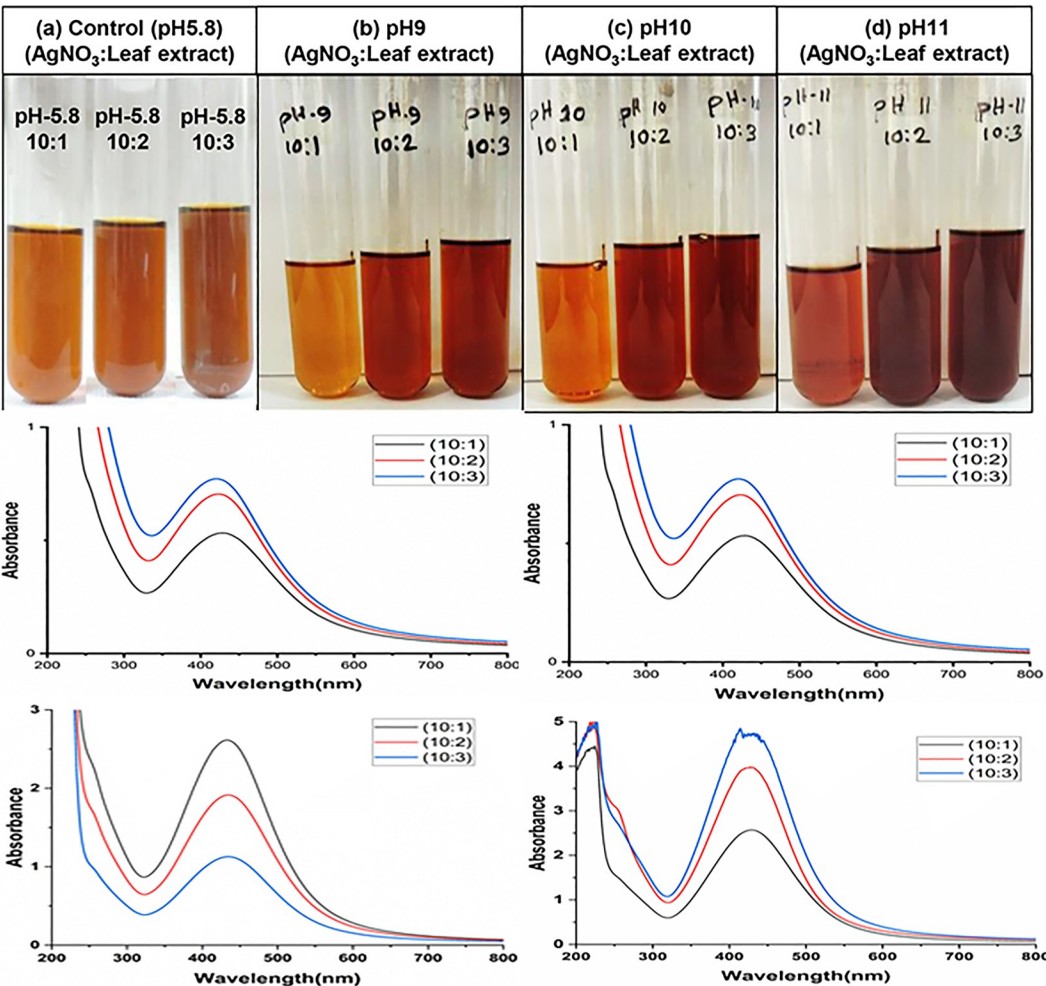

**Fig 1.** Color change after AgNPs formation (a-d) and UV-vis spectra showing absorbance with different concentrations and pH of plant extract (e-h).

of Dholkolmi leaf extract. 0.1 N KOH was used to adjust the pH of leaf extract solutions. The variation of pH was 9 to11 with an accuracy of ± 0.2 (Fig 1).

## 2.3. Characterization of AgNPs

The samples were characterized by different analytical methods using sophisticated instruments for observing the optical properties of biosynthesized nanoparticles. The samples were initially analyzed using UV–vis spectrophotometer (Model Shimadzu UV- 2600i, Japan) to detect the presence of AgNPs at a specific wavelength. To observe the crystalline and amorphous nature of AgNPs, X-ray diffraction (XRD) evaluations of AgNPs powder were carried out by Smart Lab, Rigaku, Japan. By employing Cu K radiation (0.154 nm) at a scanning rate of 30.0˚/min ranging from 5˚ to 90˚ and an X-ray tube operating at 40kV and 50mA, XRD patterns were captured. To examine whether functional groups were present on the surface of nanoparticles, Fourier Transform Infrared Spectroscopy (FTIR; Perkin Elmer, Spectrum II) was utilized. The oven-dried AgNPs powder was subjected to FTIR spectroscopy measurements. Particle size distribution of AgNPs was observed by the Dynamic Light Scattering

(DLS) method and the zeta potentials were determined using Litesizer 500 (Anton Paar, United Kingdom). Scanning Electron Microscope (SEM) observation was done by JCM7000, JEOL, Japan at the Plant Pathology Division lab of Bangladesh Rice Research Institute (BRRI), Gazipur-1701. The experiment was carried out at a 15 KV accelerating voltage. After applying a gold coating on the slide, a SEM picture was obtained. Using an EDX spectrometer, the elemental makeup of the produced nanoparticles was identified. Additionally, to capture the quality structure of AgNPs, the JEOL-7600F was used to do a Field Emission Electron Microscope (FESEM) study.

## 2.4. Antifungal activity of AgNPs

Different concentrations of AgNPs with the following ratio was prepared to test the efficacy of silver nano solutions. A total of 17 treatments i.e., $AgNO_3$:D (Dholkolmi) = 10:1, $AgNO_3$:D = 10:2, $AgNO_3$:D = 10:3, $AgNO_3$:D = 10:1(pH9), $AgNO_3$:D = 10:2(pH9), $AgNO_3$:D = 10:3 (pH9), $AgNO_3$:D = 10:1(pH10), $AgNO_3$:D = 10:2(pH10), $AgNO_3$:D = 10:3(pH10), $AgNO_3$: D = 10:1(pH11), $AgNO_3$:D = 10:2(pH11), $AgNO_3$:D = 10:3(pH11), $AgNO_3$, Amistar Top (1 ppm), Disease control, Leaf extract and Water were mixed individually with Potato Dextrose Agar (PDA) media (poisoned food method) to assess the *in vitro* antifungal effects of AgNPs against *R. solani*. In the middle of each Petri plate, 5 mm-diameter agar plugs with fungus mycelia were positioned and incubated for three days at 28°C. The growth inhibition rate of mycelia was calculated using the following equation: Inhibition rate (RH, %) = ((R-r)/R) *100 described by Khatami et al. [23], Where R is the fungal mycelia's radial growth on the control plate and r is the mycelia's radial growth on the AgNPs-treated plate. Additionally, mycelium growth was measured at four times intervals (0 hour, 24 hours, 48 hours and 72 hours) to observe the rate of fungal growth in respective of each treatment.

## 2.5. Determination of AgNPs concentration

To apply AgNPs *in vivo* experiment, AgNPs concentration was measured from the solutions of $AgNO_3$ and Dholkolmi leaf extracts. A spectrophotometer is used to determine the concentration of the prepared silver nanoparticles and the results were derived from the calibration curve based on the linear standard curve equation (y = 0.0512x + 0.0079; $R^2$ = 0.997) of standard silver nanoparticles.

## 2.6. *In vivo* examination of AgNPs on sheath blight disease under greenhouse condition

Two pot experiments were conducted to evaluate the efficacy of AgNPs against sheath blight disease. All the pots were filled with sterilized alluvial loamy soil. Thirty-day-old seedlings of highly susceptible variety BRRI dhan48 were sown in each pot to conduct the experiment. Randomized Complete Block Design (RCBD) with three replications of each treatment was followed to conduct the study. For the first pot experiment, a total of 10 treatments i.e., $AgNO_3$ (7.5 ppm), 1.5 ppm, 3 ppm, 4.5 ppm, 6 ppm, 7.5 ppm AgNPs solution, Amistar Top, Disease control, Dholkolmi leaf extracts and Healthy control were used in *in-vivo* experiment. The 3-day-old mycelial blocks of *R. solani* were placed in the center of the plants at maximum tillering stage right above the water line and observed for sheath blight disease development. The first and second sprays were applied to each treatment after two and ten days after inoculation, respectively. Sheath blight disease severity was assessed at 21 days after disease inoculation and relative lesion height was calculated using the following formula: Relative lesion height (RLH) = (Lesion height/Plant height)*100. Based on the result of the first pot experiment, the second pot experiment was setup similarly by increasing the concentration of the

treatments with an expectation of better performance of AgNPs. The treatments of the second experiment were AgNO$_3$ (37.5 ppm), 7.5 ppm, 15 ppm, 22.5 ppm, 30 ppm, 37.5 ppm of AgNPs solution, Disease control, Dholkolmi leaf extracts, and Healthy control.

## 2.7. Effect of AgNPs under field conditions on sheath blight disease of rice

The experiment was carried out in the experimental field of the Plant Pathology Division of BRRI. A sheath blight susceptible rice variety BR11 was used for the field experiment. The plot size was 3m × 3m and a randomized complete block design with three replications was used to carry out the study. Thirty-day-old seedlings were transplanted at a 20 cm to 20 cm spacing and recommended agronomic practices were maintained during the study. Rice plants were inoculated with 3-day-old culture of *R. solani* at the maximum tillering stage. Eight treatments (T1: Healthy control, T2: Disease control, T3: Leaf extract (Dholkolmi), T4: AgNO$_3$ (11.25 ppm), T5: Amistar Top (1 ppm), T6: 6 ppm AgNPs, T7: 7.5 ppm AgNPs and T8: 11.25 ppm of AgNPs solution were sprayed two times in each plot. The first spray and second spray were applied at 3 days and 10 days after inoculation, respectively.

## 3. Results

### 3.1. Biosynthesis and characterization of AgNPs

**3.1.1. Visual observation.**   The silver nitrate solution began to change color when the Dholkolmi leaf extract was added. After 30 minutes of constant stirring, the solution gradually darkened. The color of the reaction mixtures changed from light orange to reddish brown indicating the production of AgNPs in the solution (Fig 1). Leaf extract concentration and pH of the leaf extracts played a vital role in the synthesizing AgNPs. The color of the reaction becomes darkened when the pH and concentration of the leaf extract were increased (Fig 1), suggesting more production of AgNPs in the solution. After 30 min, the solution of the 3 mL leaf extract at pH11 caused a rapid change in color from light orange to dark brown, indicating the fast reduction of Ag$^+$ to Ag$^0$ in AgNO$_3$ solution, while the color in other samples was changed after incubation for 0.3–8 h in a dark condition and the control sample remained as light orange color.

**3.1.2. UV–vis absorbance studies.**   The results of the UV-visible spectrophotometer showed a spectrum of surface plasmon resonance (SRP) with an absorption band ranging from 421 to 434 nm, validated the synthesis of AgNPs in the solution (Fig 1). Indeed, a stronger and sharper absorption band at 433 nm was visible in the SPR spectra of AgNPs produced from the increased concentration of leaf extract (Fig 1H). Additionally, the UV-vis spectra results revealed that the reaction mixture's absorbance intensity increased with time, leaf extract concentration, and pH.

**3.1.3. X-ray diffraction (XRD).**   X-ray diffraction (XRD) analysis was used to analyze the AgNPs from Dholkolmi leaf extracts (Fig 2A). Four intense diffraction peaks at 2θ values of 38.04˚, 46.13˚, 64.43˚, and 76.61˚ were indexed to the (111), (200), (220) and (311) planes of silver reflections, respectively. The average crystallite sizes of the nanoparticles were calculated by using Debye-Scherrer's equation: D = Kλ/β cos θ, where, D is the estimated crystal size from XRD patterns, θ is the Bragg angle in degrees, λ is the wavelength of X-ray source used, β is the angular width at the half maximum of the diffraction peak and K is the constant of Debye-Scherrer's equation. The estimated crystalline size of the biosynthesized silver nanoparticles was ~45 nm. Moreover, the XRD patterns confirmed that the silver nanoparticles are of face-centered cubic crystalline nature.

**3.1.4. FTIR of silver nano-particles.**   Infrared spectra of biosynthesized AgNPs were determined at the range of 450–4000 cm$^{-1}$ at room temperature. Analyzing the silver

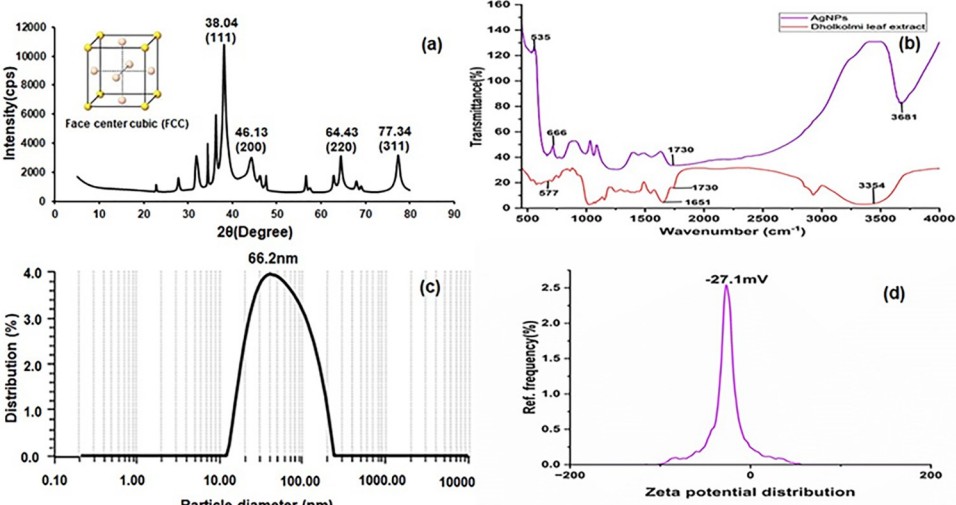

**Fig 2.** X-ray diffraction pattern (a), FTIR spectrum (b), particle size diameter (c), and Zeta potential (d) of AgNPs.

nanoparticle using FTIR made it possible to validate the existence of specific functional groups and the plant extract's dual role as a capping and reducing agent. The wave number of free hydroxyl groups and phenol groups significantly changed (3681 cm$^{-1}$), indicating that polyphenol-containing alkaloids may be involved in the reduction of silver nitrate and further stabilization of AgNPs (Fig 2B). It's interesting to note that the wave number (666 cm$^{-1}$) that corresponds to sulphonic groups also changed significantly.

**3.1.5. DLS and zeta potential.** According to the DLS pattern, this approach produces silver nanoparticles with a diameter that ranges from 12.3 nm to 110.5 nm (Fig 2C) with a polydispersity index (PDI) of 0.25. The average diameter of the particles was measured to be 66.2 nm. Zeta potential is provided as a measure of the stability of the solution; the larger the absolute value, the stronger the electrostatic repulsion. The biosynthesized AgNPs zeta potential in water was measured to check the stability of synthesized AgNPs. The zeta potential of AgNPs by Dholkolmi leaf extract in the current investigation was -27.1 mV at pH 10.0 (10:1 ratio) (Fig 2D). The formulation becomes more stable as a result of the large negative value, which reinforces the repulsion between the particles.

**3.1.6. SEM and FESEM with EDX of AgNPs.** The high-density silver nanoparticles synthesized by the *I. carnea* leaf extract were visible in the SEM and FESEM (Fig 3A and 3B), which demonstrated the polydisperse AgNPs of various sizes. The particles were spherical in shape. Strong silver signals and weak oxygen, aluminum, and carbon peaks in the EDX profile were indicative of the presence of biomolecules attached to the surface of the AgNPs (Fig 3C).

**3.1.7. TEM analysis.** The size, form, and morphology of nanoparticles have all been determined using transmission electron microscopy (TEM). The produced silver nanoparticles in the TEM micrograph were found to be correctly distributed, spherical, and ranged in size from 46.38 to 73.81 nm with a few particles being bigger than average due to nanoparticle coagulation (Fig 3D).

## 3.2. *In vitro* examination of inhibitory effects of AgNPs on mycelial growth of *R. solani*

The antifungal potential of biosynthesized silver nanoparticles was evaluated in the current study utilizing the poisoned food approach against *R. solani* and compared with the control

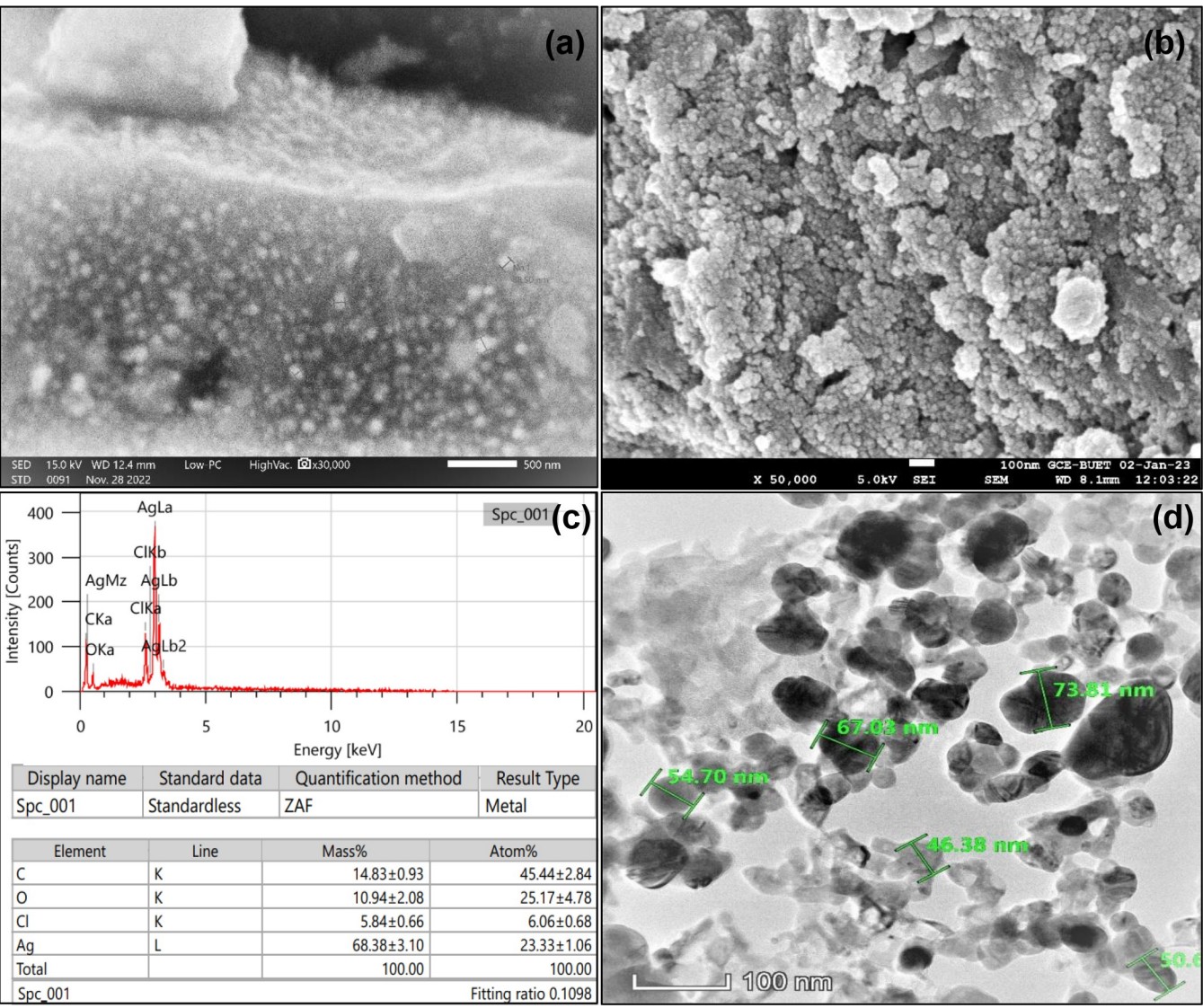

**Fig 3.** SEM (a), FESEM (b), EDX spectra (c), and TEM (d) images of biosynthesized AgNPs.

(water, AgNO$_3$ (1.5 ppm), Amistar Top (1 ppm), Leaf extract). The findings showed that the *in vitro* growth of *R. solani* was considerably suppressed by both the Amistar top (0.4 cm) and AgNO$_3$: Dholkolmi leaf extract = 10:1 (pH 10) (1.2 cm) treatments. The effect of tested AgNPs with different concentrations on mycelial growth is presented in Fig 4. Among the different treatments of Dholkolmi leaf extract (D) mediated AgNPs, Ag: D = 10:1 (pH 10) showed the most significant mycelium growth reduction (1.2 cm) over control (8.7 cm). Over time (0 hours to 72 hours) mycelium growth inhibition was significantly increased in most of the treatments (Fig 5). In every time interval, Amistar Top (91 to 97% mycelium inhibition), AgNO$_3$ (62 to 86% mycelium inhibition) and AgNO$_3$:Dholkolmi leaf extract = 10:1 (pH10) (70 to 89% mycelium inhibition) performed well compared to other treatments (Fig 5).

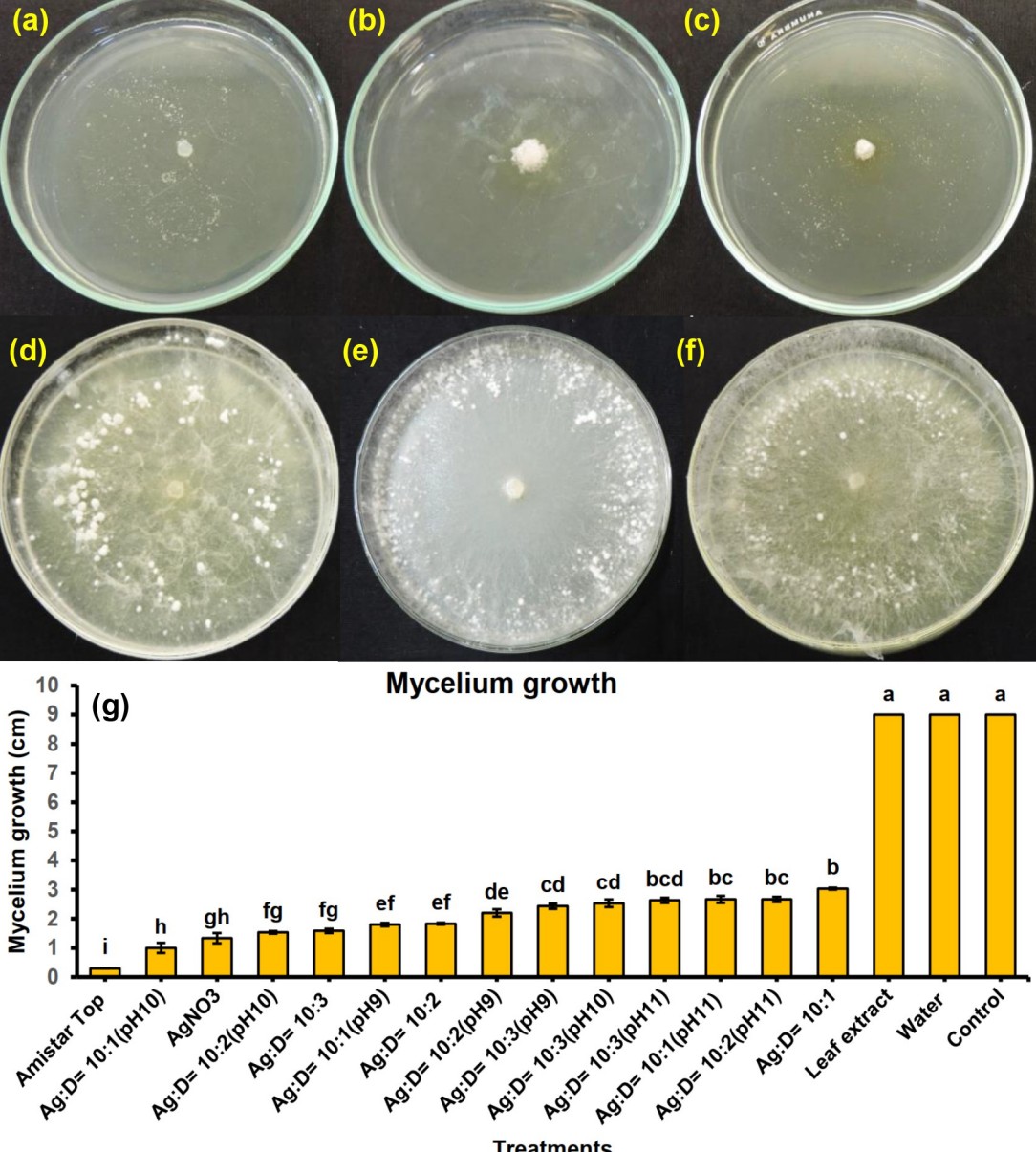

**Fig 4. Efficacy of synthesized AgNPs against *R. solani* mycelium growth inhibition in *in vitro* conditions.** Here, (a) Fungicide (Amistar Top), (b) AgNO₃, (c) AgNPs, (d) Disease control (*R. solani*), (e) Leaf extract, and (f) Water (g) effect of Dholkolmi leaf extract mediated AgNPs on mycelial growth of *R. solani*, Ag = AgNO₃, D = Dholkolmi leaves extract.

### 3.3. *In vivo* examination of AgNPs on sheath blight disease under greenhouse condition

Different concentrations of Dholkolmi leaf extract-mediated AgNPs were tested against *R. solani* for the management of sheath blight disease (Fig 6). Varying levels of inhibitory effects were found in plants treated with AgNPs. In the first pot experiment, the highest disease inhibition (15.40% RLH) was noticed in AgNPs (7.5 ppm) which was significant compared to disease control (23.12%) (Fig 6A). In the second pot experiment, a similar pattern was found and a significant reduction (13.27% RLH) was found in AgNPs (37.5 ppm) compared to disease

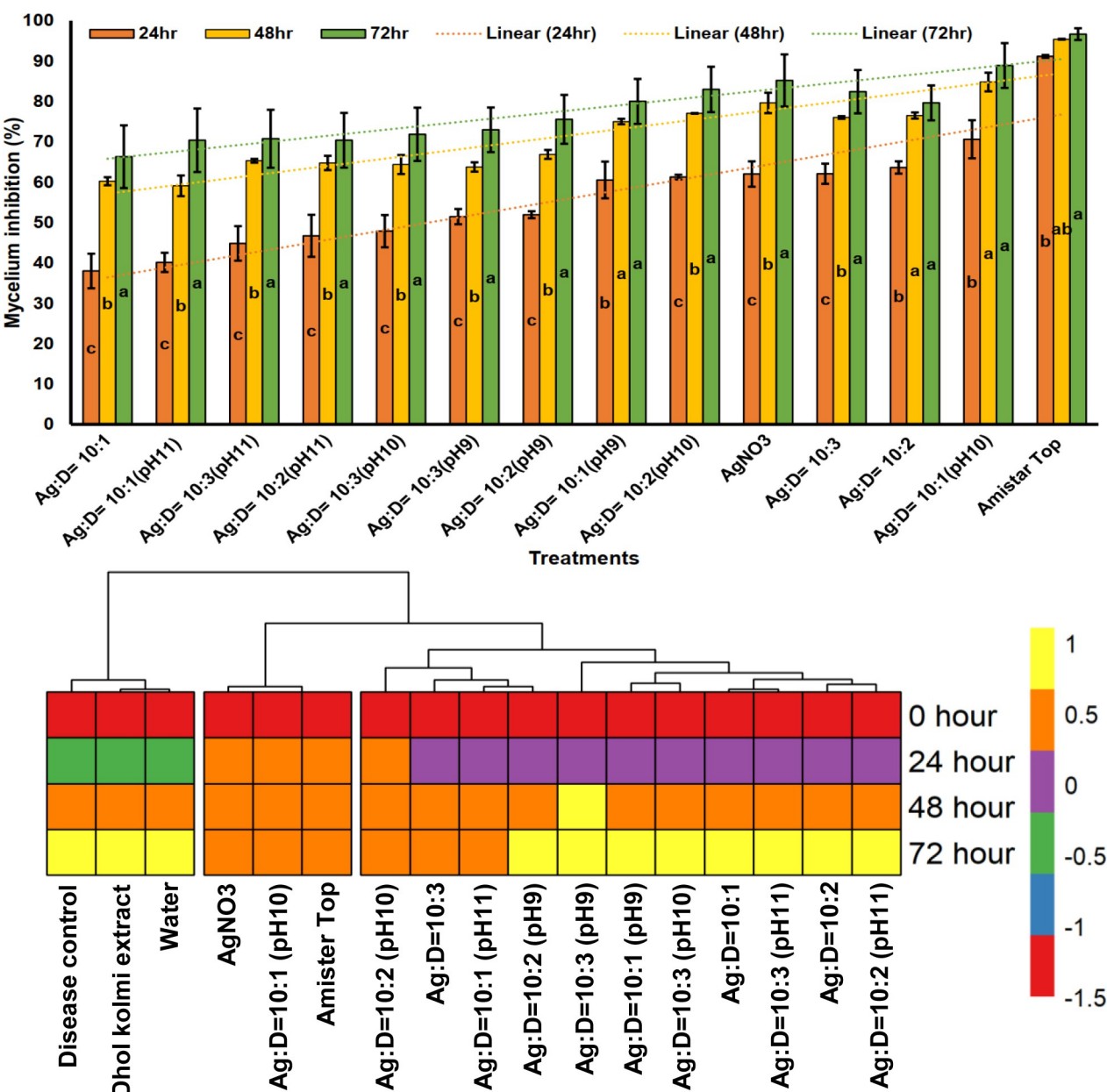

**Fig 5. Hourly efficacy of AgNPs at different concentrations and pH to control the mycelium growth of *R. solani* at *in vitro* conditions.**

control (24.98% RLH) (Fig 6B). In both pot experiments, a higher dose of AgNPs had no significant difference with Amistar Top (Fig 6A and 6B) which provides evidence that AgNPs have a potent antifungal effect to reduce sheath blight lesion in the rice tillers.

### 3.4. Effect of AgNPs on sheath blight disease under field condition

The effect of different treatments on sheath blight at field conditions are shown in Fig 6C. The highest percent of RLH was recorded in disease control (64.25%) and the lowest RLH was found in Amistar Top (23.65% RLH) followed by 11.25 ppm AgNPs (34.03% RLH), 7.5 ppm AgNPs (36.80% RLH), 6 ppm AgNPs (44.39% RLH) and 11.25 ppm AgNO$_3$ (51.92% RLH) (Fig 6C). A field reaction of AgNPs against sheath blight disease is shown in Fig 7.

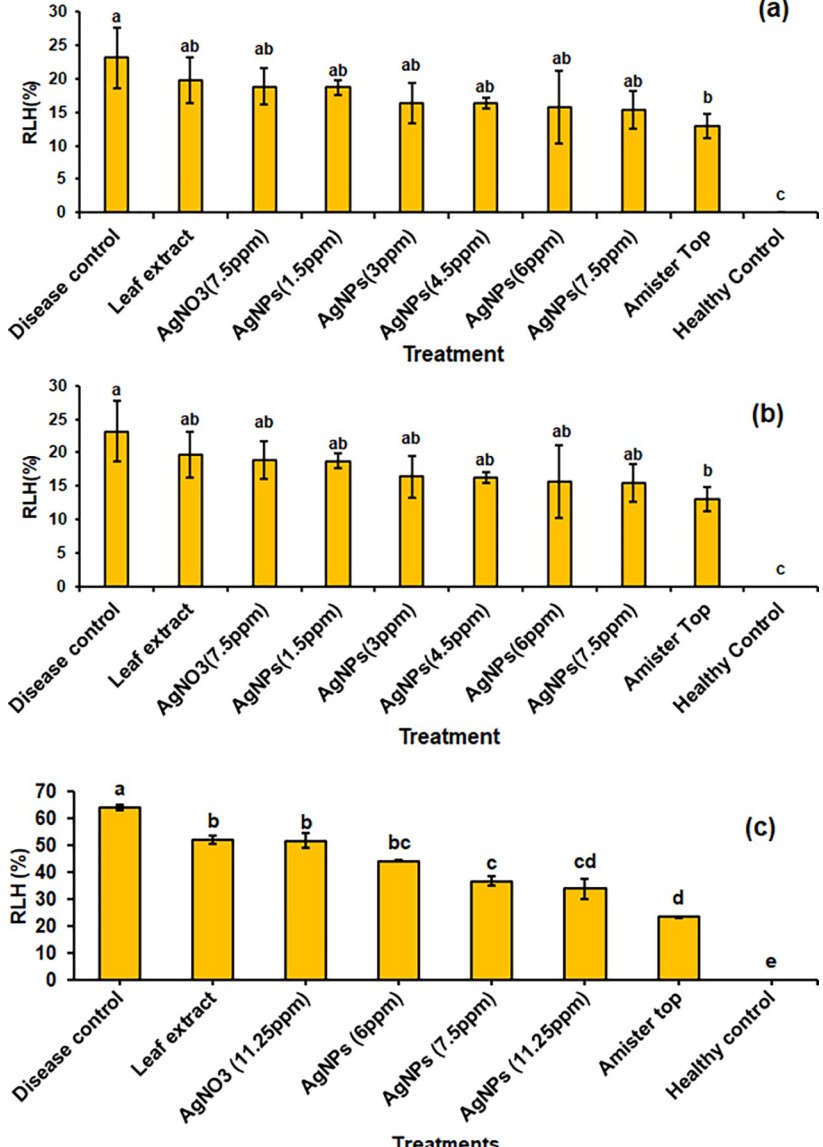

**Fig 6.** Efficacy of AgNPs against sheath blight disease of rice in a pot (a-b) and field experiments (c).

## 4. Discussion

Nanotechnology emerges as a cutting-edge approach in the quest to manage crop diseases, offering significant potential to transform agricultural practices. Despite its considerable promise, the application of nanoparticles in combating rice diseases remains relatively under-explored. Our study bridges this gap by focusing on the environmentally friendly synthesis of silver nanoparticles (AgNPs) and conducting a comprehensive characterization, followed by thorough evaluation in laboratory, greenhouse, and field trials to combat sheath blight disease in rice. The favorable results obtained from the application of AgNPs against sheath blight disease highlight the promise of nanoparticle-based strategies in disease management for rice crops. However, the effectiveness of AgNPs is dependent on the concentration used for pathogen treatment, emphasizing the critical need for dosage optimization. Our study unveiled that synthesis of AgNPs formed after mixing silver nitrate with Dholkolmi leaf extract even at

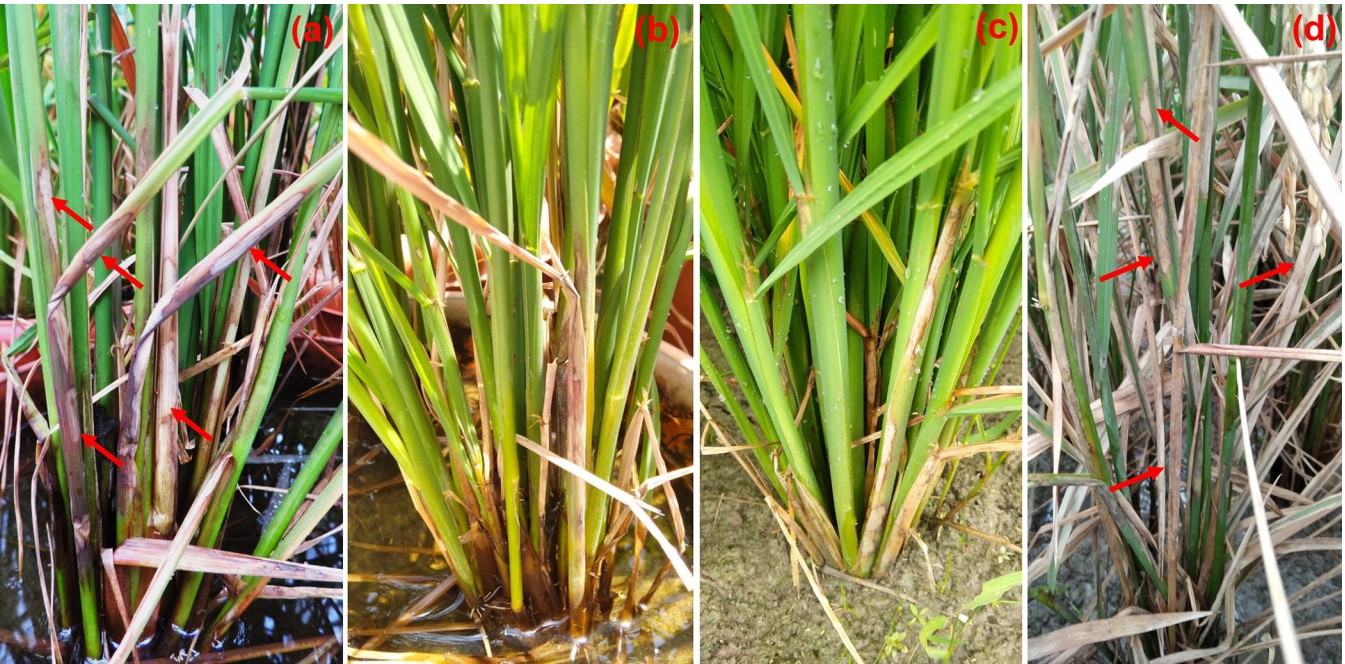

**Fig 7. Sheath blight disease suppression due to the application of AgNPs application.** Here, (a) disease control and (b) AgNPs are treated.

different ratios and hence, the change in dark brown color was observed due to the formation of nanoparticles formation, which has been further justified by ultraviolet–visible spectroscopy. Here, the intensity of the color change from light yellowish to dark brown was directly related to the quantity of the leaf extract. The reddish-brown color exhibited by AgNPs in an aqueous solution indicates the activation of surface plasmon vibrations and reduction of AgNO3 [50].

In this study, we employed UV-vis spectroscopy to observe the physical conversion of silver ions into AgNPs. Analysis of the UV-vis spectra revealed a distinct peak ranging between 421 and 434 nm, which serves as a clear indicator of silver nanoparticle production. The absorbance peak falling within the 400 to 500 nm range further corroborates the presence of AgNPs [51, 52]. The precise location of this absorption band is influenced by the medium's dielectric constant and the species adsorbed on the nanoparticle surfaces. The symmetrical nature of the band suggests uniform dispersion of nanoparticles with a spherical morphology [53]. This finding aligns with the observations of Balavijayalakshmi and Ramalakshmi, who also noted the spherical form of silver nanoparticles [54].

In addition, we observed that an increase in the pH and volume of leaf extracts led to a noticeable change in reaction color, shifting from a light orange hue to a deep brown, indicative of AgNPs production in the solutions. Both pH and the concentration of reactants play crucial roles in determining the intensity of the resulting color [44]. The shape and size of the particles were notably influenced by pH, owing to its ability to modulate the charge of biomolecules, thereby impacting their capping and stabilizing functions [55, 56]. Furthermore, we observed that pH notably expedited the reduction reaction, evident in how quickly the solution transitioned to a dark brown hue when AgNO$_3$ was combined with aqueous Dholkolmi leaf extract at pH 11. This acceleration triggered surface plasmon oscillations in silver nanoparticles, giving them a brown appearance in aqueous solution [57]. According to Khalil et al. [58], elevating the pH of the solution from 2 to 8 enhances the absorbance of AgNPs produced from olive leaf extract, with

absorbance declining as pH increases further. However, our study revealed a consistent increase in absorbance as pH rose from 9 to 11, similar findings were recorded by Vanaja [59].

XRD results revealed the presence of AgNPs having crystalline size was ~45 nm with face-centered cubic (fcc) shape, while the peaks at 2θ value of 38 degree. The XRD data were compared to the database of the Joint Committee on Powder Diffraction Standards (JCPDS, file nos. 04–0783) and the pure crystalline silver structure was noticed. These findings were consistent with previous research, where AgNPs were also identified as crystalline [60]. The synthesized AgNPs from Dholkolmi leaf extract exhibited nanocrystals arranged in a Face Center Cubic (FCC) pattern, consistent with observations by Sundeep et al. [61].

To identify the key factors for the silver ions (Ag+) reduction into AgNPs (Ag0) in the Dholkolmi leaf extract, FTIR analysis was carried out. The results confirmed that biomolecules played a vital role as reducing and capping agents of AgNPs. In the case of AgNPs, a shift in the absorbance peak with variable band intensity was observed at different points when compared to the control Dholkolmi leaf extract-based AgNPs spectra and it revealed different absorption bands ranging from 535 to 3681 cm$^{-1}$. This predicts the presence of possible biomolecules that are involved in the reduction and stabilization of silver ions (Ag+) to AgNPs (Ag0) present in aqueous leaf extract. Previous studies on the phytochemistry of Dholkolmi (*I. carnea*) have identified calystegineB2, calystegineC1, and swainsnine alkaloids as the primary chemical constituents. Consequently, polyphenols are hypothesized to play a significant role in mediating the reduction of silver nitrate to AgNPs [62]. The average diameter of the particle size was 66.2 nm in the DLS method whereas in TEM particle size ranged from 46.38 to 73.81 nm. The DLS measured size was slightly bigger as compared to the particle size measured from TEM micrographs because the DLS method measures the hydrodynamic radius [63]. EDX signals confirmed the existence of the silver element in the synthesized AgNPs with a peak optical absorption range. The carbon signal might have come from the coating material of the instrument's coating that was adsorbed, but the oxygen signal may have come from ambient oxygen from the air. On the other hand, Cl may come from the emission of X-rays from proteins and enzymes of Dholkolmi leaf extract. Previous research verified silver Surface Plasmon Resonance as an average absorption peak of about 2.5 KeV [64, 65].

In our *in vitro* study, a significant inhibition (92%) of mycelial growth of *R. solani* was found at a ratio of 10:1 of pH 10 with AgNPs (1.5 ppm, particle size ranging from 30–90 nm) compared to other concentrations. Biogenic AgNPs due to their smaller particle size and stronger capping by biomolecules, exhibit enhanced activity. These findings underscore the potential of biogenic AgNPs as effective antifungal agents. Silver nanoparticles exhibited significant antifungal properties, as evidenced by their effectiveness against *R. solani*, the causal agent of sheath blight disease in rice. Previous studies have demonstrated the efficacy of AgNPs against other rice fungi such as *Bipolaris sorokiniana* and *Magnaporthe oryzae* [66]. For instance, AgNPs with a size range of 20–30 nm inhibited the growth of *Magnaporthe grisea* by 72.8% when applied at a concentration of 200 mg/mL [66]. Nanoparticles possess high reactivity and can be effective at very low concentrations when combined with fertilizers or pesticides [67, 68]. Moreover, they are considered harmless and have been shown to enhance seed germination percentage, vigor, plant height, and reduce disease severity caused by pathogens like *Fusarium oxysporium* on various crops [69]. In specific concentrations, AgNPs have shown remarkable inhibition rates against *R. solani*. For instance, AgNPs ranging from 40–60 nm inhibited *R. solani* growth by 43.3–73.6% at a dose of 2 mg/mL [70]. Additionally, AgNPs of 10–20 nm size resulted in a 67% inhibition of *R. solani* at 100 mg/mL [71], while AgNPs with a size of 5–10 nm demonstrated an 85% inhibition at 50 mg/mL [72]. Complete suppression of *Aspergillus flavus* mycelium was achieved with AgNPs of 32.7 nm at concentrations ranging from 16–199 mm [73].

The results from the *in vivo* pot experiment indicated that a concentration of 1.5 ppm AgNPs was less effective in combating *R. solani*. However, this concentration exhibited maximum suppression of mycelial growth in the *in vitro* bioassay. Initial pot experiments revealed that the highest disease suppression was achieved at a concentration of AgNPs (7.5 ppm), although it did not reach the efficacy level of the fungicide. Consequently, the concentration of AgNPs was escalated in the subsequent pot experiment. In the second pot experiment, AgNPs at a concentration of 37.5 ppm proved to be the most effective among various concentrations tested, exhibiting a disease suppression rate of RLH 14.25% compared to the fungicide with RLH of 15.5%. Furthermore, investigations conducted in net houses demonstrated that AgNPs at 50 ppm positively impacted the fresh and dry weights of rice plants while significantly suppressing the growth of leaf lesions [74].

Nanoparticles possess a remarkable ability to combat phytopathogenic fungi, with various mechanisms elucidated by researchers to unveil their mode of action. AgNPs, for instance, target cell membranes and structural components, leading to cell denaturation. They disrupt transport mechanisms, including ion efflux, as demonstrated in prior studies [75]. Due to their high surface-to-volume ratio, both silver ions and silver nanoparticles exhibit potent antimicrobial characteristics, interacting with cell membranes and rupturing cell wall structures [76]. Researchers like McDonnell and Russell have shown that nanoparticles interact with phosphorus or sulfur, making proteins containing, such as DNA, ideal binding sites for nanoparticles [77]. The size and surface area of silver nanoparticles enable them to connect with cell DNA, facilitating interaction with microbial cell membranes and the release of silver ions, which enhance membrane permeability [78, 79]. Factors such as cell wall thickness, composition, peptidoglycan concentration, and lipopolysaccharide charge contribute to how AgNPs affect cell walls [80]. The antimicrobial actions of AgNPs primarily involve adherence to microbial cells, penetration inside cells, formation of ROS (Reactive Oxygen Species) and free radicals, and regulation of microbial signal transduction pathways [81]. Overall, AgNPs have emerged as effective agents for controlling diseases in various crops, owing to their size and efficacy. Our findings further underscore the pivotal role of AgNPs in managing plant pathogens, particularly in controlling sheath blight disease in rice.

## 5. Conclusion

The study utilized *I. carnea* (Dholkolmi) to develop a straightforward and environmentally friendly method for producing AgNPs. This synthesis process proved efficient in terms of both reaction time and nanoparticle stability, all without the need for external stabilizers or reducing agents. The biosynthesized AgNPs were confirmed using UV-visible spectrophotometer, DLS, SEM, FESEM, TEM, XRD, and Zeta potential measurements. In vitro studies demonstrated significant suppression of mycelial growth by AgNPs. Pot experiments revealed notable disease inhibition rates, with AgNPs at concentrations of 7.5 ppm and 37.5 ppm showing particularly promising results compared to disease control. In field trials, AgNPs exhibited varying degrees of disease suppression, with the lowest relative lesion height recorded in treatments involving AgNPs. Overall, the findings suggest that AgNPs have the potential to effectively manage and suppress sheath blight disease in rice, offering an eco-friendly approach to disease management.

## Supporting information

**S1 Data.**
(XLSX)

## Acknowledgments

The authors are grateful to the authority of Bangladesh Rice Research Institute (BRRI) for providing research facilities to conduct this experiment.

## Author Contributions

**Conceptualization:** Mohammad Abdul Latif.

**Data curation:** A. K. M. Sahfiqul Islam, Rejwan Bhuiyan, Mohammad Ashik Iqbal Khan.

**Formal analysis:** A. K. M. Sahfiqul Islam, Sheikh Arafat Islam Nihad.

**Funding acquisition:** Mohammad Abdul Latif.

**Investigation:** Mohammad Abdul Latif.

**Methodology:** A. K. M. Sahfiqul Islam, Rejwan Bhuiyan, Sheikh Arafat Islam Nihad, Rumana Akter, Mohammad Ashik Iqbal Khan, Shamima Akter, Md. Rashidul Islam, Md. Atiqur Rahman Khokon, Mohammad Abdul Latif.

**Project administration:** Mohammad Abdul Latif.

**Resources:** Mohammad Ashik Iqbal Khan, Mohammad Abdul Latif.

**Software:** A. K. M. Sahfiqul Islam, Sheikh Arafat Islam Nihad.

**Supervision:** Md. Rashidul Islam, Md. Atiqur Rahman Khokon, Mohammad Abdul Latif.

**Validation:** A. K. M. Sahfiqul Islam, Mohammad Abdul Latif.

**Visualization:** A. K. M. Sahfiqul Islam, Sheikh Arafat Islam Nihad, Mohammad Abdul Latif.

**Writing – original draft:** A. K. M. Sahfiqul Islam, Sheikh Arafat Islam Nihad, Mohammad Abdul Latif.

**Writing – review & editing:** A. K. M. Sahfiqul Islam, Sheikh Arafat Islam Nihad, Mohammad Abdul Latif.

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
