## [Decision Letter · Decision Letter 0]

29 Jan 2024

PONE-D-23-39382Green synthesis and characterization of silver nanoparticles and its efficacy against Rhizoctonia solani, a fungus causing sheath blight disease in ricePLOS ONE

Dear Dr. Latif,

Thank you for submitting your manuscript to PLOS ONE. After careful consideration, we feel that it has merit but does not fully meet PLOS ONE’s publication criteria as it currently stands. Therefore, we invite you to submit a revised version of the manuscript that addresses the points raised during the review process.

We look forward to receiving your revised manuscript.

Kind regards,

Marwa Fayed

Academic Editor

PLOS ONE

Journal Requirements:

"This experiment was conducted under the project titled “Sustainable management of blast, sheath blight and bacterial blight diseases of rice through Nanoparticles (Project ID:TF 71-C/20) funded by Krishi Gobeshona Foundation (KGF), Ministry of Agriculture, Bangladesh."

"Authors are grateful to authority of Plant Pathology Division of Bangladesh Rice Research Institute (BRRI) and Krishi Gobesona Foundation to provide funding and research facilities to conduct this experiment."

"This experiment was conducted under the project titled “Sustainable management of blast, sheath blight and bacterial blight diseases of rice through Nanoparticles (Project ID:TF 71-C/20) funded by Krishi Gobeshona Foundation (KGF), Ministry of Agriculture, Bangladesh."

5. We note that your Data Availability Statement is currently as follows: All relevant data are within the manuscript and its Supporting Information files.

Additional Editor Comments:

Kindly revise your manuscript extensively according to the attached reviewers' comments.

Add updated references and ensure that they follow the journal's guidelines.

Extensive language and grammar revision is required.

Reviewers' comments:

Reviewer's Responses to Questions

**Comments to the Author**

1. Is the manuscript technically sound, and do the data support the conclusions?

Reviewer #1: Partly

Reviewer #2: Partly

Reviewer #3: Yes

Reviewer #4: Yes

Reviewer #5: Yes

Reviewer #6: No

2. Has the statistical analysis been performed appropriately and rigorously? 

Reviewer #1: I Don't Know

Reviewer #2: No

Reviewer #3: Yes

Reviewer #4: Yes

Reviewer #5: Yes

Reviewer #6: No

3. Have the authors made all data underlying the findings in their manuscript fully available?

Reviewer #1: Yes

Reviewer #2: No

Reviewer #3: Yes

Reviewer #4: Yes

Reviewer #5: Yes

Reviewer #6: No

4. Is the manuscript presented in an intelligible fashion and written in standard English?

Reviewer #1: Yes

Reviewer #2: No

Reviewer #3: No

Reviewer #4: Yes

Reviewer #5: Yes

Reviewer #6: No

5. Review Comments to the Author

Reviewer #1: The manuscript entitled "Green synthesis and characterization of silver nanoparticles and its efficacy against Rhizoctonia solani, a fungus causing sheath blight disease in rice" has been reviewed.

The overall manuscript is good. However, before publishing it, there are several issues should be resolved. There are several minor grammatical errors, typo errors have been identified.

The abstract need some modification,.

Introduction is ok.

In the material and methods section, some of the methodology needs detailed description.

For better readership and improvement in the manuscript, author can read the following literature and cite them at appropriate places:

- Phytogenically Synthesized Zinc Oxide Nanoparticles (ZnO-NPs) Potentially Inhibit the Bacterial Pathogens: In Vitro Studies

- Nano-pesticidal potential of Cassia fistula (L.) leaf synthesized silver nanoparticles (Ag@CfL-NPs): Deciphering the phytopathogenic inhibition and growth augmentation in Solanum lycopersicum (L.)

- Green Synthesized Silver Nanoparticles Mitigate Biotic Stress Induced by Meloidogyne incognita in Trachyspermum ammi (L.) by Improving Growth, Biochemical, and Antioxidant Enzyme Activities

Reviewer #2: The article lacks the innovation, since similar work has been published before, please follow this link: https://doi.org/10.1049/iet-nbt.2015.0121

The English language of the manuscript is very poor. It must be improved (please see carefully the attached revised version). The manuscript is difficult to publish before the editing of English language.

The Abstract

The abstract is too long; it should be shortened and the abbreviations should not mention in the abstract. For example, the genera of organisms should be written in full.

AgNO3 correct to AgNO3 throughout the manuscript.

In Materials and Methods:

The paragraph 2.2. is confusing and must be modified for understanding.

The authors did not explain the characteristics of soil used in pot experiment.

The paragraph 2.6. is confusing and must be modified for understanding.

Line 303: What are the authors means by the 3-day old mycelial blocks R. solani were placed in the center of the plants?

Dholkolimi or dholkolimi, please unify throughout the manuscript.

Amister or amister,

In vivo or in vitro should be written in italic.

The color or colour, please unify throughout the manuscript.

Min. or minutes, please unify throughout the manuscript.

The author must follow the style of the journal during writing particularly the references list.

Reviewer #3: Green synthesis and characterization of silver nanoparticles and its efficacy against

Rhizoctonia solani, a fungus causing sheath blight disease in rice in this titled paper author expressed the viws nicely some observatons are

1. Figure 2 a make spetra in a cristanality form like 100 , 101 201 etc and have compared the data with JCPD number

2. FTIR graph overlaying with standard silve nitrate graph and check it starts from higher range to lower 4000 to 600

3. XRD shows it is not in a pure form it contains C, O Cl also and explain about atoms percentage in grpah and Ag is less

4. for all UV graphs there is a peak in UV range what it indiactes, and why u have taken control as pH 5.5

5. References are not in same style

6. have you taken any standard pesticide as control for comparision

Reviewer #4: A good manuscript with sound information. However, some improvement is needed before it gets published. The following are the recommendations:

1. Please signify the novelty of the work. Developing AgNPs via plant sources and observing their anti-microbial activity is not new. Please indicate the novelty of your work given plant used, process used, or fungi used in the exp.

2. Please mention the solution used to increase the pH of plant extract for developing AgNPs.

3. Why pH 7 and 8 are missing? Do you have any specific reason behind this?

4. FTIR data of plant extract is missing, if possible please add the same.

5. Please improve the discussion part.

Reviewer #5: The manuscript entitled:" Green synthesis and characterization of silver nanoparticles and its efficacy against Rhizoctonia solani, a fungus causing sheath blight disease in rice" has been reviewed. In my opinion, it s very interesting and falls really well with the scope of the journal. So I recommend it to be accepted.

Kind regards

Reviewer #6: The manuscript is technically week ad needs major improvements.

The figures are very poorly taken, not appropriate and unable to read. there are many statistical errors and typo mistakes. Whole manuscripts is needed to be rewrite.

6. PLOS authors have the option to publish the peer review history of their article (what does this mean?). If published, this will include your full peer review and any attached files.

Reviewer #1: **Yes: **Mohammad Shahid

Reviewer #2: No

Reviewer #3: No

Reviewer #4: No

Reviewer #5: **Yes: **Sonia Aghighi

Reviewer #6: No

---

## [Author Response · Author response to Decision Letter 0]

2 Apr 2024

Response to Editor and reviewer’s comments

Journal Requirements:

Comment 1: Please ensure that your manuscript meets PLOS ONE's style requirements, including those for file naming. The PLOS ONE style templates can be found at 

Response: We checked and tried to arrange the manuscript according to PLOS ONE journal format.

Comment 2. We suggest you thoroughly copyedit your manuscript for language usage, spelling, and grammar. 

Response: We tried to revise the language, spelling and grammer of our manuscript. The manuscript has been edited by Dr. Mohammad Abdul Latif, Director, Admin and Common Service and Former Head and Chief Scientific Officer of the Plant Pathology Division of Bangladesh Rice Research Institute (BRRI), Gazipur, Bangladesh.

Comment 3: Thank you for stating the following financial disclosure: 

"This experiment was conducted under the project titled “Sustainable management of blast, sheath blight and bacterial blight diseases of rice through Nanoparticles (Project ID:TF 71-C/20) funded by Krishi Gobeshona Foundation (KGF), Ministry of Agriculture, Bangladesh."

Response: We already included the statement "The funders had no role in study design, data collection and analysis, decision to publish, or preparation of the manuscript” in the funding section.

Comment 4. Thank you for stating the following in the Acknowledgments Section of your manuscript: 

"Authors are grateful to authority of Plant Pathology Division of Bangladesh Rice Research Institute (BRRI) and Krishi Gobesona Foundation to provide funding and research facilities to conduct this experiment."

"This experiment was conducted under the project titled “Sustainable management of blast, sheath blight and bacterial blight diseases of rice through Nanoparticles (Project ID:TF 71-C/20) funded by Krishi Gobeshona Foundation (KGF), Ministry of Agriculture, Bangladesh."

Response: We removed the funding information from the Acknowledgement section and updated the funding statement in the Funding section.

Comment 5. We note that your Data Availability Statement is currently as follows: All relevant data are within the manuscript and its Supporting Information files.

Response: We attached our data as a supporting file. Please check.

Additional Editor Comments:

Kindly revise your manuscript extensively according to the attached reviewers' comments.

Response: We tried to revise the manuscript according to reviewers’ comments.

Add updated references and ensure that they follow the journal's guidelines.

Response: Tried to update the reference and style according to journal guidelines.

Extensive language and grammar revision is required.

Response: Tried to revise the language of the whole manuscript.

Reviewers' comments:

Response to Reviewer's comments

Comments to the Author

1. Is the manuscript technically sound, and do the data support the conclusions?

Reviewer #1: Partly

Reviewer #2: Partly

Reviewer #3: Yes

Reviewer #4: Yes

Reviewer #5: Yes

Reviewer #6: No

2. Has the statistical analysis been performed appropriately and rigorously?

Reviewer #1: I Don't Know

Reviewer #2: No

Reviewer #3: Yes

Reviewer #4: Yes

Reviewer #5: Yes

Reviewer #6: No

3. Have the authors made all data underlying the findings in their manuscript fully available?

Reviewer #1: Yes

Reviewer #2: No

Reviewer #3: Yes

Reviewer #4: Yes

Reviewer #5: Yes

Reviewer #6: No

4. Is the manuscript presented in an intelligible fashion and written in standard English?

Reviewer #1: Yes

Reviewer #2: No

Reviewer #3: No

Reviewer #4: Yes

Reviewer #5: Yes

Reviewer #6: No

Reviewers Comments to the Author

Reviewer #1:

Comment 6: The manuscript entitled "Green synthesis and characterization of silver nanoparticles and its efficacy against Rhizoctonia solani, a fungus causing sheath blight disease in rice" has been reviewed.

The overall manuscript is good. However, before publishing it, there are several issues should be resolved. There are several minor grammatical errors, typo errors have been identified.

Response: Thank you for your nice comments. The grammatical errors as well as typo errors existing in the preliminary manuscript was reviewed and solved throughout the corrected manuscript.

Comment 7: The abstract needs some modification.

Response: The abstract was modified accordingly. Page: 2

Comment 8: In the material and methods section, some of the methodology needs detailed description.

Response: In the material and methods section, methodology part was revised with detail explanation especially in 2.2 and 2.6 section. Page: 5 & Page 7.

Reviewer #2:

We are very much grateful for your valuable comments. We tried to answer your queries in the following section

Comment 9: The English language of the manuscript is very poor. It must be improved.

Response: We tried to revise the language of our manuscript. The manuscript has been edited by Dr. Mohammad Abdul Latif, Director, Admin and Common Service and Former Head and Chief Scientific Officer of the Plant Pathology Division of Bangladesh Rice Research Institute (BRRI), Gazipur, Bangladesh.

Comment 10: The abstract is too long; it should be shortened and the abbreviations should not be mentioned in the abstract. For example, the genera of organisms should be written in full.

Response: The abstract was modified accordingly. Page: 2.

Comment 11: AgNO3 is correct to AgNO3 throughout the manuscript.

Response: An accurate form of AgNO3 was placed throughout the manuscript. 

Comment 12: The paragraph 2.2. is confusing and must be modified for understanding.

Response: The paragraph 2.2 was rewritten more elaboratively for clear understanding. Page: 5.

Comment 13: The authors did not explain the characteristics of the soil used in the pot experiment

Response: In the pot experiment we used sterilized alluvial loamy soil which was not reported as toxic to any crop before. However, we have a control treatment as well through which we can extract the key effect of sole Ag nanoparticles in the experiment. Page: 7.

Comment 14: The paragraph 2.6 is confusing and must be modified for understanding.

Response: The paragraph 2.6 was rewritten accordingly. Page: 7.

Comment 14: Line 303: What are the authors mean by the 3-day old mycelial blocks R. solani were placed in the center of the plants?

Response: We used 3-day-old mycelial blocks of R. solani to inoculate the rice plants. It means young culture (3 days old) of sheath blight pathogen was used for pathogenicity test. Youn culture of R. solani are more active to cause disease initiation so that the efficacy of the used nanoparticles would be more distinguishable.

Comment 15: Dholkolimi or dholkolimi, please unify throughout the manuscript.

Response: Dholkolmi is unified throughout the manuscript. Page: 2, 5, 6, 7, 8, 9, 10, 11 etc.

Comment 16: Amistar or amister?

Response: Author replaced amister with Amistar (the correct spelling and wording) throughout the manuscript.

Comment 17: In vivo or in vitro should be written in italic.

Response: Author replace both in vivo or in vitro in italic form throughout the manuscript.

Comment 18: The color or colour, please unify throughout the manuscript.

Response: Author used color instead of colour through- out the manuscript.

Comment 19: Min. or minutes, please unify throughout the manuscript.

Response: Author used minutes in throughout the manuscript. Page: 5, Page: 8.

Comment 20: The author must follow the style of the journal during writing particularly the references list.

Response: The author corrected the bibliography style according to the journal style. Page 17-25.

Reviewer #3:

Comment 21: Figure 2 a make spectra in a cristanality form like 100 ,101, 201 etc and have compared the data with JCPD number.

Response: Thank you for your valuable comments. The author corrected the Figure 2a accordingly. Please check Figure 2a.

Comment 22: FTIR graph overlaying with standard silver nitrate graph and check it starts from higher range to lower 4000 to 600.

Response: Author modified the FTIR graph with standard silver nanoparticles along with leaf extracts. Here, the range exists between 4000-450 cm-1. (Figure 2 b).

Comment 23: XRD shows it is not in a pure form it contains C, O, Cl also and explain about atoms percentage in graph and Ag is less.

Response: EDX peaks revealed C, O, Al and may be due to the carbon-coated copper grid with aluminum stub, and Cl might be come from leaf extract.

The weight percentage of an element is the weight of that element measured in the sample divided by the weight of all elements in the sample multiplied by 100.

The atomic percentage is the number of atoms of that element, at that weight percentage, divided by the total number of atoms in the sample multiplied by 100.

So, the atomic weight percent is calculated from the element weight percentage by dividing each element weight percentage by its atomic weight, do this for all elements in the sample, you will have a list of atomic proportions. Sum these together to obtain a total atomic weight. Then for each element in the sample divide its atomic proportion by the total and * 100.

Here is an example -. EDX will give you

C: el wt% 14 O: el wt% 10.9 Cl: el wt% 5.84 Ag: el wt% 68.38

divide each by their atomic weights C (12), O (16), Cl (35.5), Ag (107.868) and sum

C: 14 / 12 = 1.16 O: 10.9/16 = 0.68 Cl: 5.84/35.5 = 0.16 Ag: 68.38/ 107.868= 0.63

divide each by the sum (2.63) and turn into atomic percentage

Ag: 0.63/ 2.63 * 100 = 24.1(23.33±1.06) at atomic wt%

Comment 24: for all UV graphs there is a peak in UV range what it indicates, and why u have taken control as pH 5.5

Response: UV- vis spectrometer was used for the primary characterization of nanoparticles. This peak indicates the presence of AgNPs. In this study, the effect of pH was studied in two different conditions including fresh leaf extract (pH-5.8) and alkaline conditions.

Comment 25: References are not in same style.

Response: The author corrected the bibliography style according to the journal style. Page 17-25.

Comment 26: have you taken any standard pesticide as a control for comparison?

Response: The author took a standard pesticide (Amistar top) as a control for comparison.

Reviewer #4: 

A good manuscript with sound information. However, some improvement is needed before it gets published. 

Response: We are very much grateful for your valuable comments. We tried to answer your queries in the following section

Comment 27: Please signify the novelty of the work. Developing AgNPs via plant sources and observing their anti-microbial activity is not new. Please indicate the novelty of your work given plant used, process used, or fungi used in the exp.

Response: In this study, antifungal efficacy was evaluated against R. solani using Dholkolmi mediated AgNPs. AgNPs were also evaluated under field conditions as liquid form. Dholkolmi mediated synthesis of Ag nanoparticles and their characterization using different method and testing the efficacy of synthesized AgNPs against rice sheath blight pathogen are the novelty of our works. 

Comment 28: Please mention the solution used to increase the pH of plant extract for developing AgNPs.

Response: The author used KOH to increase the pH of plant extract for developing AgNPs. Page: 5; Line- 153.

Comment 29: Why pH 7 and 8 are missing? Do you have any specific reason behind this?

Response: The effect of pH was studied in two different conditions including fresh leaf extract (pH-5.8) and alkaline conditions (pH-9, 10 & 11). We have no specific reason to avoid pH 7 and 8 because we know that higher pH influences the chemical reaction compared to a neutral pH range like 6 to 8,so we directly used pH 9, 10 and 11.

Comment 30: FTIR data of plant extract is missing, if possible, please add the same.

Response: The author included FTIR data of plant extract along with AgNPs in Figure-2b.

Comment 30: Please improve the discussion part.

Response: We tried to improve the discussion of our manuscript. The discussion has been edited by Dr. Mohammad Abdul Latif, Director, Admin and Common Service and Former Head and Chief Scientific Officer of the Plant Pathology Division of Bangladesh Rice Research Institute (BRRI), Gazipur, Bangladesh.

Reviewer #5: 

The manuscript entitled:" Green synthesis and characterization of silver nanoparticles and

---

## [Decision Letter · Decision Letter 1]

20 May 2024

Green synthesis and characterization of silver nanoparticles and its efficacy against Rhizoctonia solani, a fungus causing sheath blight disease in rice

PONE-D-23-39382R1

Dear Dr. Abdul Latif,

We’re pleased to inform you that your manuscript has been judged scientifically suitable for publication and will be formally accepted for publication once it meets all outstanding technical requirements.

Kind regards,

Marwa Fayed

Academic Editor

PLOS ONE

Additional Editor Comments (optional):

Reviewers' comments:

Reviewer's Responses to Questions

**Comments to the Author**

Reviewer #1: All comments have been addressed

Reviewer #2: All comments have been addressed

Reviewer #4: All comments have been addressed

2. Is the manuscript technically sound, and do the data support the conclusions?

Reviewer #1: Yes

Reviewer #2: Partly

Reviewer #4: Yes

3. Has the statistical analysis been performed appropriately and rigorously? 

Reviewer #1: Yes

Reviewer #2: Yes

Reviewer #4: Yes

4. Have the authors made all data underlying the findings in their manuscript fully available?

Reviewer #1: Yes

Reviewer #2: Yes

Reviewer #4: Yes

5. Is the manuscript presented in an intelligible fashion and written in standard English?

Reviewer #1: Yes

Reviewer #2: Yes

Reviewer #4: Yes

6. Review Comments to the Author

Reviewer #1: The authors have improved the manuscript as suggested by the reviewer. Now the manuscript sounds good. It can be published in its current form.

Reviewer #2: I think the authors did all requested comments

I have no research ethics

I have no publications ethics

Reviewer #4: (No Response)

7. PLOS authors have the option to publish the peer review history of their article (what does this mean?). If published, this will include your full peer review and any attached files.

Reviewer #1: **Yes: **Mohammad Shahid

Reviewer #2: No

Reviewer #4: **Yes: **Dr. Deepak Gola

---

## [Editor Report · Acceptance letter]

27 May 2024

PONE-D-23-39382R1 

PLOS ONE

Dear Dr. Latif, 

I'm pleased to inform you that your manuscript has been deemed suitable for publication in PLOS ONE. Congratulations! Your manuscript is now being handed over to our production team.

Kind regards, 

on behalf of

Prof. Marwa Fayed 

Academic Editor

PLOS ONE